# The Effect of In Situ Synthesis of MgO Nanoparticles on the Thermal Properties of Ternary Nitrate

**DOI:** 10.3390/ma14195737

**Published:** 2021-10-01

**Authors:** Zhiyu Tong, Linfeng Li, Yuanyuan Li, Qingmeng Wang, Xiaomin Cheng

**Affiliations:** 1School of Materials Science and Engineering, Wuhan University of Technology, Wuhan 470070, China; tongzy@whut.edu.cn (Z.T.); lilinfeng0514@whut.edu.cn (L.L.); yyli@whut.edu.cn (Y.L.); 2School of Electromechanical and Automobile Engineering, Huanggang Normal University, Huanggang 438000, China; wangqingmeng@whut.edu.cn

**Keywords:** MgO nanoparticles, eutectic nitrates, in situ, specific heat capacity, thermal diffusion coefficient

## Abstract

The multiple eutectic nitrates with a low melting point are widely used in the field of solar thermal utilization due to their good thermophysical properties. The addition of nanoparticles can improve the heat transfer and heat storage performance of nitrate. This article explored the effect of MgO nanoparticles on the thermal properties of ternary eutectic nitrates. As a result of the decomposition reaction of the Mg(OH)_2_ precursor at high temperature, MgO nanoparticles were synthesized in situ in the LiNO_3_–NaNO_3_–KNO_3_ ternary eutectic nitrate system. XRD and Raman results showed that MgO nanoparticles were successfully synthesized in situ in the ternary nitrate system. SEM and EDS results showed no obvious agglomeration. The specific heat capacity of the modified salt is significantly increased. When the content of MgO nanoparticles is 2 wt %, the specific heat of the modified salt in the solid phase and the specific heat in the liquid phase increased by 51.54% and 44.50%, respectively. The heat transfer performance of the modified salt is also significantly improved. When the content of MgO nanoparticles is 5 wt %, the thermal diffusion coefficient of the modified salt is increased by 39.3%. This study also discussed the enhancement mechanism of the specific heat capacity of the molten salt by the nanoparticles mainly due to the higher specific surface energy of MgO and the semi-solid layer that formed between the MgO nanoparticles and the molten salt.

## 1. Introduction

The exploitation of solar energy is essential to sustainable development. To solve the problem of intermittent solar energy in solar thermal utilization, it is necessary to use the thermal energy storage (TES) system to store and release heat when solar radiation is weak or absent [1,2]. As an excellent heat storage carrier, molten salt heat storage material has the advantages of sizeable latent heat, high energy storage density, low subcooling, good thermal stability and low cost, which is widely used as a heat storage medium for solar heat. Currently, solar salt is widely used in TES technology. The components of solar salt are NaNO_3_ and KNO_3_. The melting point of solar salt is 220 °C. Due to its high melting point, the pipeline needs to be heated to a higher temperature to prevent the pipeline from freezing, resulting in additional energy input and power generation costs. The higher melting point of molten salt limits the application of molten salt in the field of heat storage. Therefore, the ideal heat transfer fluid is supposed to have a low melting point, reducing the risk of freezing and heating energy consumption of the pipeline. Studies have shown that mixing several molten salts in a certain proportion forms a eutectic salt that can reduce the melting point while ensuring the thermal stability of the molten salt. The main melting salts are nitrates, carbonates, and sulfates [3,4]. In recent years, the development of low-melting, high-stability multi-element molten salt systems has become a research hotspot in molten salt modification. Wu et al. [5] formulate 19 kinds of binary mixed molten salts in different proportions, the main component of which is KNO_3_-Ca(NO_3_)_2_·4H_2_O. The results showed that the thermodynamic properties of these molten salts performed well. Ren et al. [6] further explored the Ca(NO_3_)_2_–NaNO_3_ binary salt and modified it with expanded graphite, which effectively improved the thermophysical properties of the molten salt. In recent years, more and more ternary and quaternary molten salts have been developed [7,8,9,10,11,12,13]. The main research systems are LiNO_3_–NaNO_3_–KNO_3_, NaNO_3_–NaNO_2_–KNO_3_, Ca(NO_3_)_2_–NaNO_3_–KNO_3_, and LiNO_3_–NaNO_3_–KNO_3_–Ca(NO_3_)_2_. These multi-molten salts have lower melting points and higher stability. The use of relevant phase diagrams to create ternary or higher salt mixtures can obtain low melting point molten salts. The ideal freezing temperature for Hitec and Hitec XL is 120–140 °C, and they can withstand temperatures exceeding 500 °C. The LiNO_3_–NaNO_3_–KNO_3_ ternary mixture is considered as a promising heat transfer and storage medium, with a low melting point (120 °C) and high thermal stability (550 °C). Multi-element eutectic molten salt has a wide operating temperature range (low melting point and high decomposition point), which is very suitable as a heat transfer fluid and heat storage carrier in the TES system of a concentrating solar power plant to store solar energy.

Nanomaterials have special physical and chemical properties due to their unique structure, so they have important applications in heat storage [14,15,16,17,18]. For example, nano-SiC and nano-MgO have not only higher specific heat capacity but also better heat transfer efficiency, and they are very good heat storage materials. Therefore, the research of nanomaterials is of great significance to the development of heat storage materials. Researchers tried to add nanoparticles to molten salt to increase the specific heat capacity of molten salt. Among the research of using nanoparticles to modify molten salt, the most common materials are SiO_2_ and Al_2_O_3_ nanoparticles, most of which have been observed to have an increase in the specific heat and thermal conductivity [19,20,21]. Dudda et al. [22] and Seo et al. [23] explored the effect of nanoparticle size on the specific heat capacity of the nanoparticle/molten salt eutectic mixture. It was observed that the salt compounds around the nanoparticles formed a large number of nano-sized structures, which may be the main reason for the increase in specific heat. From the view of structure, one reason for the increased specific heat is the thermal resistance of the interface between the nanoparticles and the molten salt. Another reason is that a semi-solid layer is formed between the nanoparticles and the molten salt. From the perspective of energy, the high surface energy of nanoparticles can also store part of the thermal energy. Hu et al. [24] performed molecular dynamics simulations on Al_2_O_3_ nanoparticles doped in solar salt and explored the reason for the specific heat enhancement from the view of energy. The result shows that the change of Coulomb energy is the reason for the change of specific heat capacity.

The addition of nanoparticles can also improve the heat transfer performance of molten salt to a certain extent. Gupta et al. [25] added different types of nanoparticles (TiO_2_, ZnO, Fe_2_O_3_, and SiO_2_) to the phase change material (PCM) of Mg(NO_3_)_2_·6H_2_O and formed the PCM–metal oxide nanocomposite material through the melting and mixing technology. The PCM–metal oxide nanocomposite with a 0.5 wt % nanoparticle addition increased the thermal conductivity by 147.5% (TiO_2_), 62.5% (ZnO), 55% (Fe_2_O_3_), and 45% (SiO_2_), respectively. Ho et al. [26] discussed the effect of nanoparticle concentration on the convective heat transfer performance of molten nano-HITEC fluid laminar flow in microtubes. The heat transfer performance of HITEC fluid with Al_2_O_3_ nanoparticle concentration as high as 0.25 wt % has been improved. The study of Yu et al. [27] observed that SiO_2_ and TiO_2_ nanoparticles can improve the thermal conductivity of molten salt. Under normal circumstances, the thermal conductivity of molten salt is about 0.2–2.0 W/(m·K), and the specific heat is about 1.35 J/(g·°C). The specific heat of the molten salt added with these two kinds of nanoparticles increased by 28.1%, and the thermal conductivity increased by 53.7%. Studies have shown that there are heat transfer channels in high-density nanostructures, which can contribute to the enhancement of thermal conductivity. D. Shin et al. [28] found that in traditional nanofluids, nanoparticles can form fractal fluid nanostructures to enhance thermal conductivity.

Poor particle dispersion can reduce the specific heat capacity of the molten salt [29]. Therefore, to achieve the particular heat enhancement of nanomaterials, the preparation method must be carefully controlled. There are many methods of using nanoparticles to modify molten salts, such as the high-temperature melting method, aqueous solution method, combustion method, and in situ synthesis method. The high-temperature melting method is to directly melt and stir molten salt and nanoparticles at high temperature to form a uniform eutectic system. The aqueous solution method is to dissolve the molten salt in water, then add nanomaterials to form a stable suspension, and finally, by heating, precipitation to obtain the eutectic salt. The combustion method is to mix the precursor, molten salt, and fuel together, then ignite the fuel and generate a lot of heat through violent combustion, so that the molten salt forms a eutectic system. The in situ synthesis method is to mix the precursor and molten salt, and then, the precursor reacts in the molten salt at a certain temperature to generate nanoparticles. Li et al. [20] and Zhang et al. [30] used SiO_2_ and Al_2_O_3_ nanoparticles as additives and added the nanoparticles to the molten salt by the high-temperature melting method, and they successfully prepared the modified salt. Xiong et al. [21] used the aqueous solution method to prepare the SiO_2_/molten salt nanofluid successfully. Lasfargues et al. [31,32] used copper sulfate pentahydrate and titanium sulfate as precursors to synthesize CuO and TiO_2_ nanoparticles in situ in solar salt. The specific heat of solar salt was observed to increase. In our previous research, we successfully synthesized MgO nanoparticles in situ in solar salt, which significantly increased the specific heat capacity of solar salt [33].

At present, there are relatively few studies on the performance improvement of multi-element molten salt by nanoparticles [19]. In the research of nanoparticle modification of molten salt, MgO nanoparticles are an excellent modified particle. MgO has several types of bulk intrinsic defects, including oxygen and magnesium vacancies, interstitials, their agglomerates, etc. [34,35]. This has aroused the interest of many researchers. In this work, we prepared a ternary eutectic nitrate and applied the in situ generation method to generate MgO nanoparticles in molten salt. By testing the specific heat capacity, latent heat of phase change, and thermal conductivity of the prepared nitrate-based composite materials, the influence of MgO nanoparticles on the heat transfer and heat storage performance of ternary nitrate was studied.

## 2. Materials and Methods

### 2.1. Materials

Mg(OH)_2_ was obtained from Aladdin Chemical Co., Ltd., Shanghai, China. KNO_3_, NaNO_3_ and LiNO_3_ were commercially supplied by Sinopharm Chemical Reagent Co., Ltd., Shanghai, China. All chemicals are of analytical grade.

### 2.2. Preparation

The first step is the preparation of lithium nitrate–potassium nitrate–sodium nitrate eutectic nitrate. Many scholars have studied the ternary eutectic point of the LiNO_3_–NaNO_3_–KNO_3_ ternary system. Zhong et al. [36] predicted the phase diagram of the LiNO_3_–NaNO_3_–KNO_3_ ternary system and experimentally verified the predicted ternary invariant points. This result is similar to the report by Coscia et al. [37]. The mass ratio of the LiNO_3_–NaNO_3_–KNO_3_ ternary system is 29:58:13 (wt %). The ternary nitrate with the mass balance was ground in a mortar for 1 h; then, it was transferred to a crucible and placed in a resistance furnace at 300 °C for 5 h. Then, we took out the crucible, cooled it, and ground it to obtain ternary eutectic nitrate.

According to the ratio in Table 1, the magnesium hydroxide precursor was added to the ternary eutectic nitrate. Then, we ground it in an agate mortar for 30 min. After that, the mixture was transferred to a ceramic crucible and placed in a resistance furnace at 400 °C for 2 h to ensure complete decomposition of the magnesium hydroxide precursor. After taking it out, it was quickly cooled in air and ground to obtain a sample. The process is shown in Figure 1.

To further test the material, it is necessary to obtain the nanoparticles synthesized in situ. The modified salt was washed with deionized water and then centrifuged. After the nitrate is washed away, the nanoparticles are dispersed by ultrasound. Finally, the resulting nanoparticles are dried.

### 2.3. Characterization

The crystal phases of all samples were characterized by X-ray diffraction (XRD, Empyrean, PANalytical B.V., Amsterdem, Netherlands). RENISHAW Raman microscope (Raman, In Via, RENISHAW, England) was used to measure modified salts and magnesium oxide nanoparticles. We observed the microstructure of the sample with a field emission scanning electron microscope (SEM, S-4800, HITACHI, Tokyo, Japan). The component analysis of the prepared modified nitrate was examined by X-ray energy-dispersive spectroscopy (EDS, AMETEK, Berwyn, PA, USA) combined with scanning electron microscopy under a constant nitrogen flow from 30 to 200 °C at a heating rate of 10 °C/min. The specific heat capacity and latent heat of phase change of the samples were measured with a differential scanning calorimeter (DSC8500, PERKINELMER, Waltham, MA, USA). Then, we used an infrared thermal imager (TESTO-872, Testo SE&Co. KGaA, Titisee-Neustadt, Germany) to characterize the heat transfer performance of ternary nitrate modified with different percentages of MgO nanoparticles. Thermal diffusivity was obtained by a laser thermal conductivity meter (LFA457, NETZSCH, Selb, Germany).

## 3. Results and Discussions

### 3.1. Components of Modified Salt

Figure 2 shows the XRD spectra of samples S_0_, S_4_, and the product after centrifugation. There are characteristic peaks at 19.03°, 23.52°, 33.06°, 33.84°, 41.15°, and 46.61° in sample S_0_, corresponding to the (110), (111), (200), (112), (221), and (113) crystal planes of KNO_3_. Two peaks at 29.41° and 38.99° represent the (104) and (113) crystal planes of NaNO_3_, respectively. In addition, the crystal plane (104) of LiNO_3_ could be found according to the characteristic peak at 32.21°. It can be seen that the ternary nitrate was successfully prepared. After centrifugation, characteristic peaks can be seen at 36.89°, 42.86°, 62.22°, 74.58°, and 78.51°, corresponding to (111), (200), (220), (311), and (222) crystal planes of MgO. Sample S_4_ possesses several characteristic peaks of MgO at 42.96° and 62.22°. It can be seen that MgO is formed in situ in the ternary nitrate system. The characteristic peaks at 18.59°, 32.84°, 38.02°, 50.85°, 58.64°, 68.87°, and 72.03° mark to (001), (100), (101), (102), (110), (200), and (201) of Mg(OH)_2_ planes. However, no characteristic peaks of Mg(OH)_2_ were found in samples S_0_ and S_4_, indicating that the magnesium hydroxide precursor was completely decomposed when the nanoparticles were generated in situ. The reason why the characteristic peaks of Mg(OH)_2_ can be seen in the product after centrifugation might be because MgO combines with water during the centrifugation process to form a trace amount of Mg(OH)_2_.

Figure 3 shows the Raman spectra of samples S_0_, S_4_, and the product after centrifugation. In samples S_0_ and S_4_, the peak at 713 cm^−1^ corresponds to the NO_3_^−^ in-plane bending vibration (710–740 cm^−1^), and the peak at the frequency of 1049 cm^−1^ corresponds to the NO_3_^−^ symmetric stretching vibration (1020–1060 cm^−1^). In the Raman spectrum of sample S_4_ and the MgO nanoparticles obtained after centrifugation, peaks with frequencies of 1499 cm^−1^ and 1936 cm^−1^ can be observed. Combined with XRD analysis, it can be further known that MgO nanoparticles were successfully generated in situ in the ternary nitrate system [38].

### 3.2. The Structure of Modified Salt

Figure 4 is SEM photos of samples S_0_, S_2_, S_4_, S_6_, and S_7_. Figure 4a is the SEM diagram of LiNO_3_–NaNO_3_–KNO_3_ ternary nitrate. The surface of the nitrate is relatively flat, and the material is uniform. The mass fractions of MgO nanoparticles in Figure 4b–d are 1 wt %, 2 wt %, and 3 wt %, respectively. It can be seen that the MgO nanoparticles are relatively evenly dispersed among the nitrates, which shows that the nanoparticles generated in situ have good dispersibility. Figure 4e,f are SEM of modified nitrate with 5 wt % MgO nanoparticles at different magnifications. When the content reached 5 wt %, the nanoparticles had obvious agglomeration. Due to the high surface energy of nanoparticles, they will agglomerate together and deposit in the nanofluid, resulting in poor system stability. In the modified salt, the size of the nanoparticles synthesized in situ is concentrated in the range of 50–200 nm.

The element distribution of the sample was determined by the area scanning method of the energy spectrum. Figure 5 is the element distribution of modified nitrate. Figure 5c is the distribution of the Mg element. Figure 5d shows the distribution of the N element, which represents the distribution of nitrate. It can be seen from the element distribution diagram that MgO nanoparticles are evenly dispersed among the ternary nitrates.

### 3.3. Specific Heat Capacity

Figure 6a,b are the specific heat capacity curves at 50–80 °C (solid phase) and 150–200 °C (liquid phase), respectively. The overall sensible heat capacity can be judged by selecting the specific heat capacities at 60 °C (solid phase) and 170 °C (liquid phase). Table 2 shows the specific heat capacities of ternary nitrate and modified salt in the solid phase and liquid phase. For the same sample, the specific heat in the liquid state is improved compared with that in the solid state. The reason for this phenomenon is that in the molten state, the ions in the molten salt perform randomly free movement, which can carry more energy. The specific heat in the solid state increases first and then decreases with the growth of nanoparticles. When the content of nanoparticles is 2%, the solid specific heat is the largest, which is 1.479 J/(g·°C). Compared with LiNO_3_–NaNO_3_–KNO_3_ ternary nitrate, it has increased by 51.54%. With the increase of nanoparticles, the specific heat in the liquid state shows an increasing trend at the first stage and then a decreasing tendency at the following stage. When the content of nanoparticles is 2%, the specific heat of the material at the liquid state reaches the peak, which is 1.878 J/(g·°C): an increase by 44.50%.

For MgO nanoparticles to increase the specific heat capacity of nitric acid ternary salt, the first possible reason is that MgO nanoparticles have higher specific surface energy. The second reason is the interface thermal resistance between MgO nanoparticles and nitrate, which can store and release additional energy. Another reason is that the MgO nanoparticles and the surrounding molten salt form a semi-solid layer. Hu et al. [24] explained the specific heat enhancement from the perspective of Coulomb energy. Molecular dynamics simulation is used to analyze the influence of nanoparticles on the energy composition of each atom type. The results show that the change in the Coulomb energy of each atom contributes the most to the enhanced specific heat capacity.

### 3.4. Latent Heat

The DSC endothermic and exothermic curves of ternary nitrate and modified nitrate are shown in Figure 7. The endothermic peak of the melting process and the exothermic peak of the solidification process can be seen in the figure. Table 3 shows the latent heat of phase change, the onset temperature, and the melting temperature of the ternary nitrate and the modified nitrate. The phase transition temperature of different samples is unchanged basically, indicating that the addition of trace nanoparticles has little effect on the phase transition process of the material. Compared with ternary nitrate, the latent heat of nitrate doped with nanoparticles is slightly lower. On the whole, there is little difference in the latent heat of phase change. In the field of medium and low-temperature energy storage, the latent heat of phase change materials is about 130 J/g, so this modified salt can meet the requirements.

### 3.5. Heat Transfer Characteristics

Ternary nitrate has good heat transfer performance and can be used as a heat transfer fluid. Figure 8 shows the thermal diffusivity of samples S_0_, S_2_, S_4_, S_6_, and S_7_. Obviously, with the increase of MgO nanoparticles, the thermal diffusion coefficient increases significantly. When the mass fraction of MgO nanoparticles is 5%, the thermal diffusion coefficient is 0.425 mm^2^·s^−1^, which is 39.3% higher than that of the eutectic salt without MgO nanoparticles. It can be seen that the interface thermal resistance effect between nanoparticles and nitrate does not reduce the overall heat transfer performance of the material. The main reason for improving the heat transfer performance of the modified salt might be due to the high thermal conductivity of the MgO nanoparticles.

To explore the heat transfer performance of the modified nitrate, the samples S_0_, S_2_, S_4_, and S_6_ were placed on a heating plate, and an infrared thermal imaging instrument was used to characterize the heat transfer performance of the material. Figure 9 shows the infrared thermal images of samples S_0_, S_2_, S_4_, and S_6_ heated on the heating plate for different times. The heat is transferred upward from the bottom end. The heat transfer rate from sample S_0_ to sample S_6_ shows an increasing trend, and the transfer rate of sample S_6_ is the fastest at the same time. Plot the temperature of samples S_0_, S_2_, S_4_, and S_6_ at the same geometric position with heating time, as shown in Figure 10. It can be seen that the heat transfer performance of the material increases with the increase of MgO nanoparticles. The doping of nanoparticles enables the material to transfer heat in the solid state quickly, accelerating the heat transfer process of the material, thus shortening the time required for the phase change of the material, and more efficiently storing thermal energy. At the same time, when the material acts as a nanofluid in a liquid state, the heat transfer performance is greatly improved, and the heat transfer process is accelerated. The results show that modified nitrate has high sensible heat and latent heat, a suitable phase transition temperature, and high thermal conductivity, which can improve the energy storage efficiency and can be used in thermal energy storage systems.

## 4. Conclusions

1. We successfully synthesized magnesium oxide nanoparticles in situ in the ternary nitrate system through the high-temperature decomposition of the Mg(OH)_2_ precursor. When the content of MgO nanoparticles does not exceed 3 wt %, the dispersion performance is more appropriate, and there is no agglomeration. The size of the nanoparticles synthesized in situ is concentrated in the range of 50–200 nm.

2. Nano-magnesium oxide can improve the heat storage performance of nitrate. When the content of MgO nanoparticles is 2 wt %, the specific heat capacity of the solid increases by 51.54%, and the specific heat capacity of the liquid increases by 44.50%. The increase in specific heat capacity is related to the interface thermal resistance and the semi-solid layer.

3. MgO nanoparticles synthesized in situ significantly improve the heat transfer performance of ternary nitrate. When the content of MgO nanoparticles is 5 wt %, the thermal diffusion coefficient of the modified salt increased by 39.3%. In the molten state, MgO nanoparticles and ternary nitrate form a heat transfer fluid. Molten salt nanofluid is an excellent heat exchange medium in the field of heat storage, with a high heat exchange capacity.

## Figures and Tables

**Figure 1 materials-14-05737-f001:**
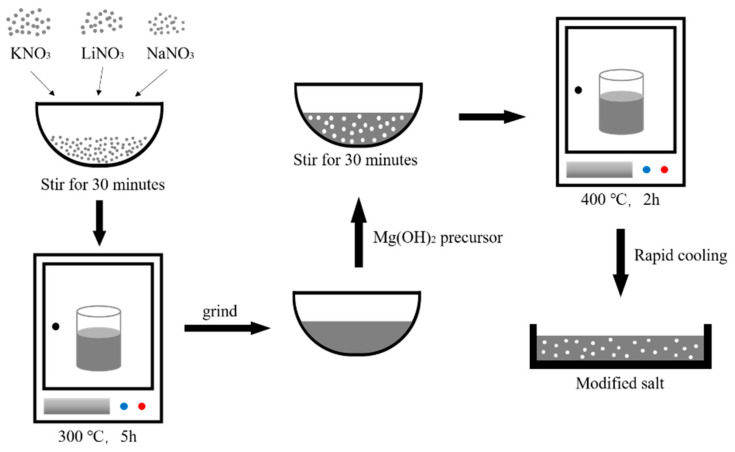
Preparation of modified salt.

**Figure 2 materials-14-05737-f002:**
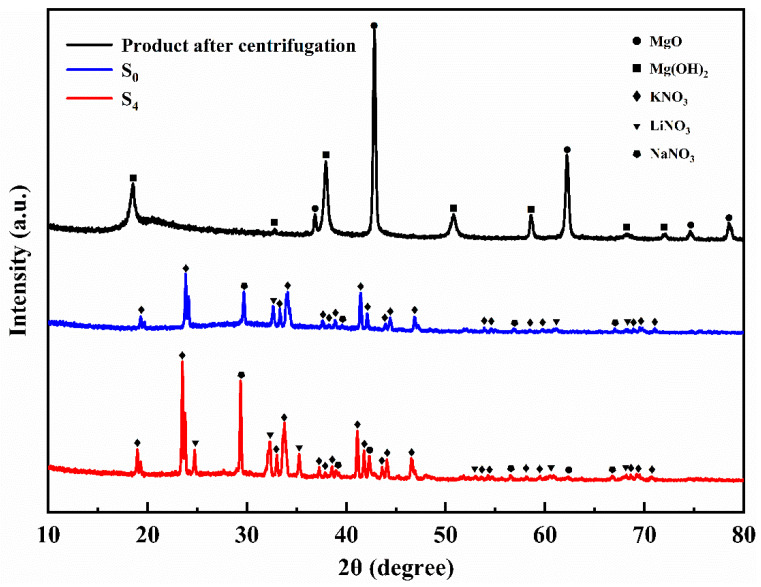
The XRD patterns of samples S_0_, S_4_, and product after centrifugation.

**Figure 3 materials-14-05737-f003:**
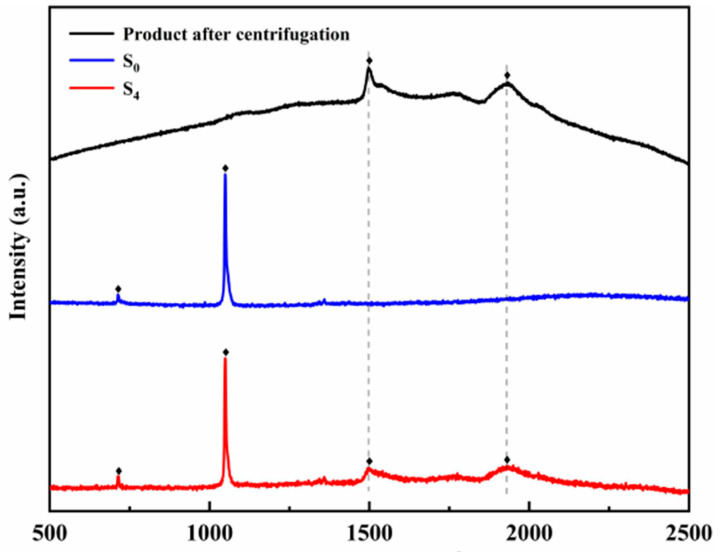
The Raman spectra of samples S_0_, S_4_, and product after centrifugation.

**Figure 4 materials-14-05737-f004:**
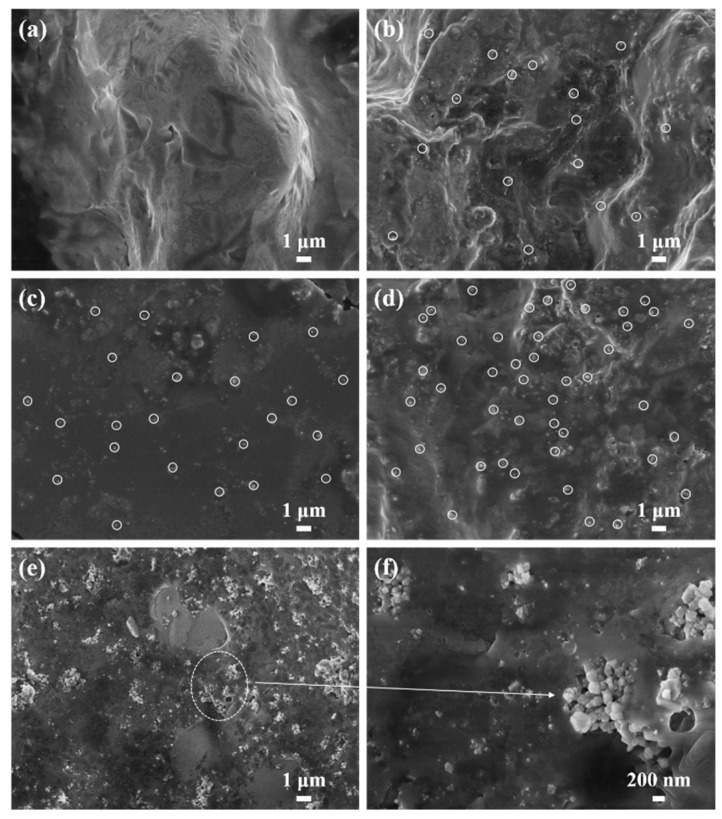
SEM micrographs of (**a**) sample S_0_, (**b**) sample S_2_, (**c**) sample S_4_, (**d**) sample S_6_, and (**e**,**f**) sample S_7_.

**Figure 5 materials-14-05737-f005:**
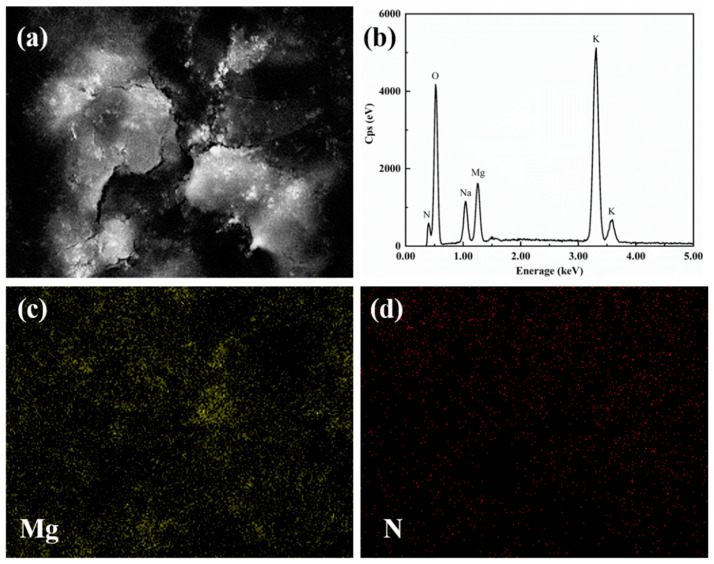
The elemental distribution of modified nitrate.

**Figure 6 materials-14-05737-f006:**
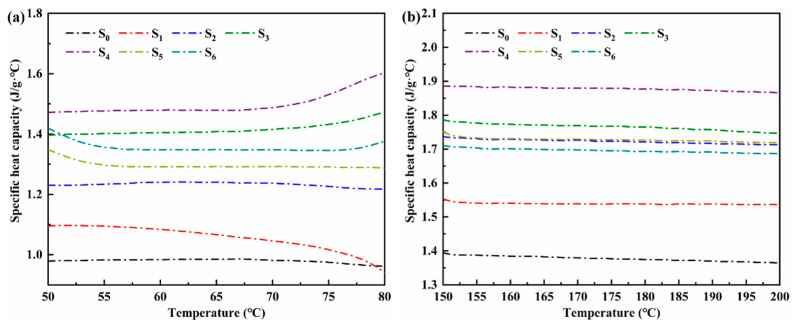
Specific heat capacity of ternary nitrate and modified nitrate: (**a**) solid phase, (**b**) liquid phase.

**Figure 7 materials-14-05737-f007:**
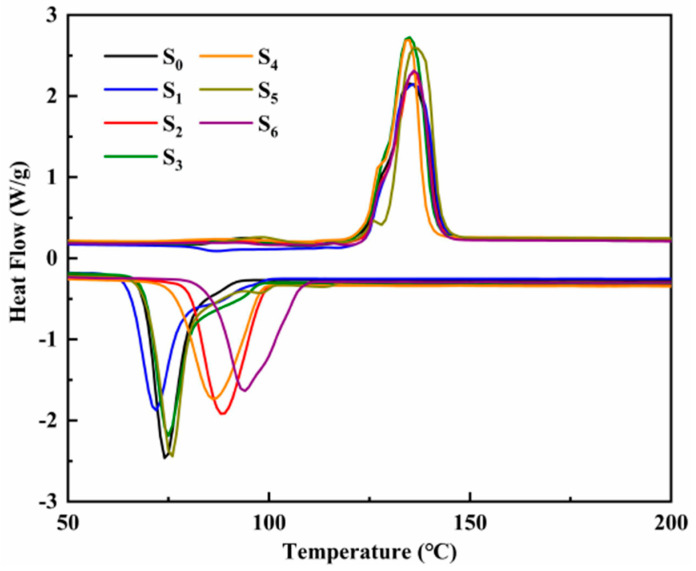
The DSC endothermic and exothermic curves of ternary nitrate and modified nitrate.

**Figure 8 materials-14-05737-f008:**
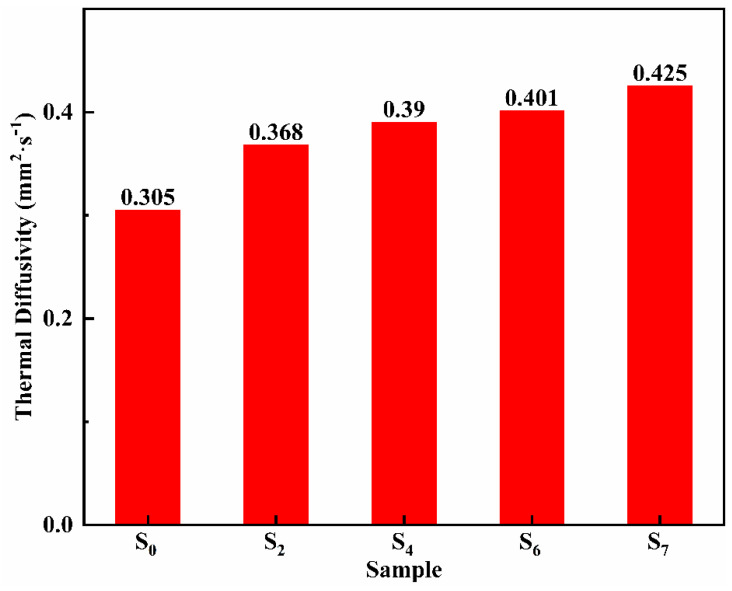
Thermal diffusivity of S_0_, S_2_, S_4_, S_6_, and S_7_.

**Figure 9 materials-14-05737-f009:**
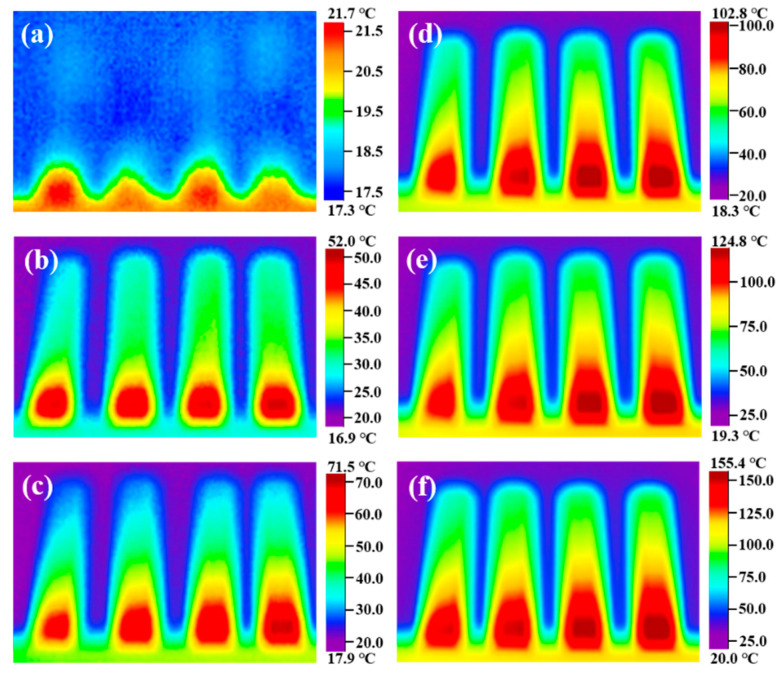
Infrared thermography images of samples S_0_, S_2_, S_4_, and S_6_ heated on the heating plate for different times: (**a**) 0 s, (**b**) 60 s, (**c**) 120 s, (**d**) 180 s, (**e**) 240 s, and (**f**) 300 s.

**Figure 10 materials-14-05737-f010:**
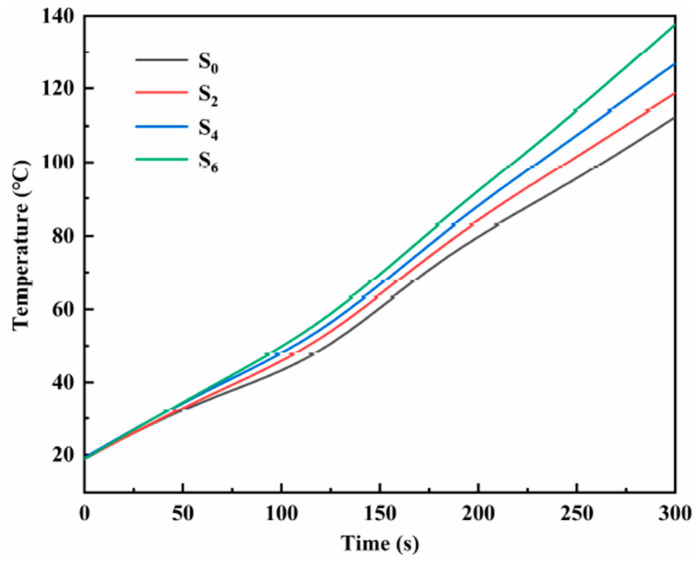
The temperature variation curve of the same geometric position of samples S_0_, S_2_, S_4_, and S_6_ with heating time.

**Table 1 materials-14-05737-t001:** The content of each ingredient in the preparation.

Sample	Percentage of MgO (wt %)	Percentage of Precursor (wt %)	Mg(OH)_2_ Precursor (g)	LiNO_3_-NaNO_3_-KNO_3_ (g)
S_1_	0.5	0.7	0.037	4.975
S_2_	1.0	1.4	0.073	4.950
S_3_	1.5	2.1	0.109	4.925
S_4_	2.0	2.8	0.145	4.900
S_5_	2.5	3.5	0.182	4.875
S_6_	3.0	4.1	0.217	4.850
S_7_	5.0	7.0	0.370	4.750

**Table 2 materials-14-05737-t002:** Specific heat capacity of ternary nitrate and modified nitrate.

Sample	Cp (J/g·°C)	
	Solid	Liquid
S_0_	0.976 (0%)	1.301 (0%)
S_1_	1.084 (11.07%)	1.538 (18.22%)
S_2_	1.240 (27.05%)	1.726 (32.67%)
S_3_	1.406 (44.06%)	1.764 (35.59%)
S_4_	1.479 (51.54%)	1.880 (44.50%)
S_5_	1.291 (32.27%)	1.727 (32.74%)
S_6_	1.348 (38.11%)	1.695 (30.28%)
Solar Salt	1.290	1.350

**Table 3 materials-14-05737-t003:** Result of latent heat, onset temperature, and melting temperature.

Sample	Latent Heat (J/g)	Onset Temperature (°C)	Melting Temperature (°C)
S_1_	135.8	110	150
S_2_	142.5	112	148
S_3_	132.3	114	152
S_4_	131.2	112	152
S_5_	138.8	116	151
S_6_	128.2	110	151

## Data Availability

All the data are available within the manuscript.

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
