# Peer review of "The Effect of In Situ Synthesis of MgO Nanoparticles on the Thermal Properties of Ternary Nitrate"

_materials, 2021, doi:10.3390/ma14195737_

Round 1

Reviewer 1 Report

Comments appear as comments in the attached pdf

Author Response

Reviewer #1

  1. What are the compositions/systems of commercial solar salt?

Response: We thank the Editor and Reviewers for these insightful comments. The components of solar salt are NaNO3 (60%) and KNO3 (40%). I have modified it and added an explanation about solar salt (line 37).

  1. What is the range of temperature of commercial salts nowadays in use?

Response: Thank you for the advice. The melting point of solar salt is 220 ℃. It still has good thermal stability at 600 °C, and will begin to decompose when it exceeds 600 °C. This content has been supplemented in the article (line 39).

  1. What molten salts? Specify, please. Add references to studies that justify this question.

Response: We thank the Editor and Reviewers for these constructive suggestions. The main melting salts are nitrates, carbonates and sulfates. The content has been revised in the text and relevant documents have been attached (line 47).

  1. In what system did they work? (line 46) Please detail they main characteristics.

Response: I am sorry for not being able to elaborate on this system, which has now been revised in the text. Wu et al. formulate 19 kinds of binary mixed molten salts in different proportions, the main component of which is KNO3-Ca(NO3)2·4H2O (line 50).

  1. With respect to what system the properties improved?

Response: In the literature mentioned, researchers have used expanded graphite to improve the performance of molten salt. Ren et al. further explored the Ca(NO3)2–NaNO3 binary salt and modified it with expanded graphite, which effectively improved the thermophysical properties of the molten salt. I have corrected it in the text (line 53).

  1. In what systems are researchers nowadys working in?

Response: Thank you for the advice. The main research systems are LiNO3–NaNO3–KNO3, NaNO3–NaNO2–KNO3, Ca(NO3)2–NaNO3–KNO3 and LiNO3–NaNO3–KNO3–Ca(NO3)2. I have corrected it in the text (line 56).

  1. What range?

Response: I apologize for not being able to give its scope. Regarding this range, the specific use temperature range of different molten salts is different. I mainly hope that the data in the previous sentence shows that this range is relatively large.

  1. Please provide examples.

Response: Thank you for the advice. For example, nano-SiC and nano-MgO not only have higher specific heat capacity, but also have better heat transfer efficiency, and are very good heat storage materials. I have corrected it in the text (line 67).

  1. What binary system?

Response: I am sorry for not being able to elaborate, this nitrate system is solar salt. I have corrected it in the text (line 84).

  1. With 0.5 wt. % four different results were obtained, please clarify this point.

Response: Thank you for the advice. The four different results correspond to the four types of nanoparticles. Gupta et al. added different types of nanoparticles (TiO2, ZnO, Fe2O3, and SiO2) to the phase change material (PCM) of Mg(NO3)2·6H2O, and formed the PCM-metal oxide nanocomposite material through the melting and mixing technology. The PCM-metal oxide nanocomposite with a 0.5 wt.% nanoparticle addition increased the thermal conductivity by 147.5% (TiO2), 62.5% (ZnO), 55% (Fe2O3), and 45% (SiO2), respectively. I have corrected it in the text (line 92).

  1. Please indicate what are the values expected for the molten salts parameters, for instance melting point, thermal conductivity, specific heat, etc., to compare with those presented in the manuscript.

Response: Thank you for the advice. I have corrected it in the text (line 98). Under normal circumstances, the thermal conductivity of molten salt is about 0.2-2.0 W/(m·K), and the specific heat is about 1.35 J/(g·°C). The specific heat of the molten salt added with these two kinds of nanoparticles increased by 28.1%, and the thermal conductivity increased by 53.7%.

  1. Please provide a short explanation about the methods.

Response: Thank you for the advice. I have corrected it in the text (line 109). The high-temperature melting method is to directly melt and stir molten salt and nanoparticles at high temperature to form a uniform eutectic system. The aqueous solution method is to dissolve the molten salt in water, then add nanomaterials to form a stable suspension, and finally, by heating, precipitation to obtain the eutectic salt. The combustion method is to mix the precursor, molten salt and fuel together, then ignite the fuel, and generate a lot of heat through violent combustion, so that the molten salt forms a eutectic system. The in-situ synthesis method is to mix the precursor and molten salt, and then the precursor reacts in the molten salt at a certain temperature to generate nanoparticles.

  1. If there are few studies, please provide references of these studies.

Response: Thank you for the advice. Related references have been added to the article.

  1. It could be interesting to indicate with arrow or circles what are the nanoparticles in SEM images.

Response: Thank you for the advice. I have used arrows or circles to indicate what are the nanoparticles in the SEM image. And replace the figure with the revised figure.

  1. It seems that there is not homogenous distribution of the particles, please clarify.

Response: Thank you for your question. The particle distribution does not seem to be uniform at the geometric angle, but when the MgO content is small, the particles do not agglomerate, indicating that the MgO dispersibility is better. Compared with the direct melting method and the aqueous solution method, the dispersibility of the in-situ synthesis method has been greatly improved. The agglomeration in Figure 4(e) is due to the excessive content of nanoparticles added. When the content is lower than this, no obvious agglomeration effect is found.

  1. What is the melting point of the proposed system? It is not clear when the transity solid-liquid take place and why you take these ranges.

Response: Since this section discusses specific heat capacity and does not involve latent heat, the melting point of the material is not mentioned. The melting point of this material is 120 °C, which will be discussed further in (3.4. Latent heat) below. The solid phase (50 °C -80 °C) and liquid phase (150 °C -200 °C) are selected because the specific heat is stable in the solid phase and the liquid phase. These two temperatures can also reflect the sensible heat capacity of the material in the two states.

  1. Figures are not in the same scale and for that reason, it is not clear.

Response: Thank you for your question. This is to explain the reason for the different specific heats of the same sample in solid and liquid conditions. Figure 6 (a) and (b) are measurements of different temperature ranges for the same batch of samples.

  1. Please clarify these values because you indicate in two cases that is the highest value.

Response: I'm sorry for not clarifying the situation here, this data is the specific heat value in liquid state. When the content of nanoparticles is 2%, the specific heat of the material in the liquid state reaches to the peak, which is 1.878 J/(g·°C), an increase by 44.50%. I have corrected it in the text (line 247).

  1. Can you provide the values of the specific surface energy?

Response: Thank you for the advice. At present, there is no test method to test it, and no specific specific surface energy value can be obtained by consulting related literature.

  1. Add reference.

Response: Thank you for the advice. I have attached references.

  1. 21. What are those requirements? please indicate.

Response: On the whole, there is little difference in the latent heat of phase change. In the field of medium and low temperature energy storage, the latent heat of phase change materials is about 130 J/g, so this modified salt can meet the requirements.

  1. Why is it obvious?

Response: Thank you for your question. Figure 8 shows the thermal diffusivity of samples S0, S2, S4, S6, and S7. The content of MgO in the samples increases successively, and its thermal diffusivity also shows an upward trend.

Reviewer 2 Report

Although this manuscript looks quite good, it is nevertheless not entirely convincing, especially from the point of view of reading its introduction.  

1. The first paragraph contains many convincing sentences, but no supporting references. Therefore, it is unclear whether this is the author's reasoning, or whether it was said by someone else, but the authors forgot to quote him .

2. Although the article is about MgO, the mention of the MgO office can only be found on page 3, line 104, where there is a short mention that "in our previous research..."  It can be seen that the authors do not report anything about the previously known data on MgО nanoparticles, although their optical properties in wide UV range  should be discussed. See for example, well-known paper: A.I. Popov et al “Comparative study of the luminescence properties of macro- and nanocrystalline MgO using synchrotron radiation” Nuclear Instruments and Methods in Physics Research Section B:   Volume 310, 1 September 2013, Pages 23-26; https://doi.org/10.1016/j.nimb.2013.05.017

as well as  González et al Phys. Rev. B 59, 4786 (1999) and references cited it.

3.   Data on Figure 3 data and related information (lines 175-185) need more corresponding discussions and sapporting references.  Note that , properties of MgO and Mg(OH)2 are expected to be different and show different optical properties. See similar situation in CdI2 (Low Temperature Physics 42, 594 (2016); https://doi.org/10.1063/1.4959019.  )

4.  Line 234.  There is no reference in "Hu explained" .... Where ? In which paper?

5. It will be also very useful if obtained data on specific heat (Table 2) will be compared with known data for the same bulk materials.

6.  Please note (line 295), that 200 nm - is already macro-sized object,

In conclusion, this paper could be recommended for publication after answering/discussion above questions/comments.   

Author Response

Reviewer #2

  1. The first paragraph contains many convincing sentences, but no supporting references. Therefore, it is unclear whether this is the author's reasoning, or whether it was said by someone else, but the authors forgot to quote him.

Response: We thank the Editor and Reviewers for these insightful comments. Many parts of this part have been further explained and related references have been added. Among them, the solar salt is further explained, and the performance of the solar salt is detailed. The other mentioned melting salt systems are also specifically explained, and relevant references are attached.

  1. Although the article is about MgO, the mention of the MgO office can only be found on page 3, line 104, where there is a short mention that "in our previous research..." It can be seen that the authors do not report anything about the previously known data on MgО nanoparticles, although their optical properties in wide UV range should be discussed. See for example, well-known paper: A.I. Popov et al “Comparative study of the luminescence properties of macro- and nanocrystalline MgO using synchrotron radiation” Nuclear Instruments and Methods in Physics Research Section B: Volume 310, 1 September 2013, Pages 23-26; as well as González et al Phys. Rev. B 59, 4786 (1999) and references cited it.

Response: Thanks for your advice. I have carefully read the two articles you recommended, and I have a more intuitive understanding of the properties of MgO, especially the optical properties. In this article, we mainly use the high thermal conductivity of MgO and the high surface energy of MgO at the nanometer scale to improve the thermophysical properties of molten salt. I have modified this part of my article to give a brief overview of the performance of MgO. As for the specific content, I have cited the two references you recommended.

  1. Data on Figure 3 data and related information (lines 175-185) need more corresponding discussions and sapporting references. Note that , properties of MgO and Mg(OH)2 are expected to be different and show different optical properties. See similar situation in CdI2 (Low Temperature Physics 42, 594 2016)

Response: Thanks for your advice. Research on the structure of cadmium oxide and cadmium hydroxide may be useful for the analysis of magnesium oxide and magnesium hydroxide. Therefore, I quoted your recommended literature. As there are few articles related to Raman spectroscopy analysis of magnesium oxide and magnesium hydroxide, I have not found the corresponding references. Regarding the composition, the material is mainly analyzed in detail through XRD. The analysis results of Raman spectroscopy are also partly combined with the XRD analysis in the previous section.

  1. Line 234. There is no reference in "Hu explained" .... Where? In which paper?

Response: We thank the Editor and Reviewers for these insightful comments. This part of the content has been modified. This part of the content is based on references, and specific documents have been inserted.

  1. It will be also very useful if obtained data on specific heat (Table 2) will be compared with known data for the same bulk materials.

Response: Thanks for your advice. I added the specific heat data of solar salt to the table to compare with the experimental data.

  1. Please note (line 295), that 200 nm - is already macro-sized object.

Response: Thank you for your question. During the synthesis process, if the size of the nanoparticles can be smaller without agglomeration, there will be better thermal performance. Although some of the synthesized nanoparticles have reached 200nm in size, most of the nanoparticles are still smaller than 100nm in size. The materials prepared at present are still stable, and the specific heat capacity and thermal conductivity of the eutectic salt are significantly improved.

Round 2

Reviewer 2 Report

This article has been significantly improved by the authors, all recommendations are taken into account, so that manuscript can be accepted and published in the "Materials".